

# Quantifying the effects of delisting wolves after the first state began lethal management

Adrian Treves*, Francisco J. Santiago-Ávila* and Karann Putrevu

Nelson Institute for Environmental Studies, University of Wisconsin, Madison, Wisconsin, United States
* These authors contributed equally to this work.

## ABSTRACT

Predators and their protection are controversial worldwide. Gray wolves, *Canis lupus*, lost U.S. federal protection (delisting) and the State of Wisconsin began lethal management first among all states and tribes that regained authority over wolves. Here we evaluated the initial success of reaching the state's explicit objective, "…to allow for a sustainable harvest that neither increases nor decreases the state's wolf population…" We used official state figures for hunter-killed wolves, population estimates from April 2017–2020, and the latest peer-reviewed model of individual wolf survival to estimate additional deaths resulting from federal delisting. More than half of the additional deaths were predicted to be cryptic poaching under the assumption that this period resembled past periods of liberalized wolf-killing in Wisconsin. We used a precautionary approach to construct three conservative scenarios to predict the current status of this wolf population and a minimum estimate of population decline since April 2020. From our scenarios that vary in growth rates and additional mortality estimates, we expect a maximum of 695–751 wolves to be alive in Wisconsin by 15 April 2021, a minimum 27–33% decline in the preceding 12 months. This contradicts the state expectation of no change in the population size. We draw a conclusion about the adequacy of regulatory mechanisms under state control of wolves and discuss the particular governance conditions met in Wisconsin. We recommend greater rigor and independent review of the science used by agencies to plan wolf hunting quotas and methods. We recommend clearer division of duties between state wildlife agencies, legislatures, and courts. We recommend federal governments reconsider the practice of sudden deregulation of wolf management and instead recommend they consider protecting predators as non-game or transition more slowly to subnational authority, to avoid the need for emergency relisting.

Corresponding author
Adrian Treves, atreves@wisc.edu

## INTRODUCTION

Wolves and their protection are controversial worldwide and across the U.S. (*Bruskotter et al., 2018*; *Manfredo et al., 2020*; *Treves & Martin, 2011*; *Chapron et al., 2014*; *Dressel, Sandström & Ericsson, 2014*). The U.S. Endangered Species Act (ESA) aims to remove listed species (delist) from federal protection once recovered but contingent on adequate regulations in subnational jurisdictions to keep them off the federal list. Two U.S. Presidential Administrations have proposed the removal of federal protections for gray wolves (*Canis lupus*) nationwide but faced dissent by majorities (if not unanimity) of their official panels of scientists (*NCEAS, 2014*; *Atkins, 2019*). The Trump administration went ahead anyway and announced on 3 November 2020 it would transfer authority to states and tribes on 3 November 2020, declaring gray wolves recovered across most of the country under the Endangered Species Act, ESA (*USFWS, 2020*). That decision asserts that the species met the criteria of the five-factor analysis (ESA 16 USC § 1531 Sec. 4(a)) among others. The five factors necessary for delisting altogether ensure the delisted species remains secure for the foreseeable future. One of those criteria is the adequacy of state and tribal (subnational) regulatory mechanisms (*Zellmer, Panarella & Wood, 2020*; *Erickson, 2012*).

Whether delisted wolves are being managed with adequate regulatory mechanisms by subnational jurisdictions seems in part a scientific question (as opposed to a values-based question), because the adequacy of the mechanisms depends on their effectiveness in regulating factors that might reverse conditions and endanger wolves again. Chief among those factors for wolves has been human-caused mortality in five U.S. wolf populations, since modern monitoring (*Treves et al., 2017*), as in other regions (*Chapron et al., 2014*; *Boitani, 1995*). We present a data point to support scientific evaluations of the adequacy of regulatory mechanisms in subnational jurisdictions, for the first state to implement recreational hunting in the wake of federal wolf delisting announced on 3 November 2020.

The State of Wisconsin wolf policy and management between 2020–2021 offers an interesting case study for the following reasons. Wisconsin was the first subnational jurisdiction to resume lethal management of wolves after delisting. The State wildlife agency (Department of Natural Resources, DNR, Annapolis, MD, USA) was explicit about its goals for regulated wolf-hunting, "The quota's objective is to allow for a sustainable harvest that neither increases nor decreases the state's wolf population…" (https://dnr.wisconsin.gov/topic/hunt/wolf/index.html, accessed 14 April 2021) and similar statements to media before the wolf-hunt (*Anderson, 2021*). There are two phrases and two parts of that objective that can be evaluated scientifically, that of "a sustainable harvest" and "neither increases nor decreases the state's wolf population". This language mirrors recent reviews of the topic that have estimated the average expected, threshold rate of human-caused mortality predicted to result in stability of wolf populations (i.e., no increase or decrease).

The estimates of stabilizing levels of human-induced mortality that would be sustainable ranges from 28–29% (*Adams et al., 2008*) to 5–10% lower estimates by (*Fuller*

*et al., 2003*; *Creel & Rotella, 2010*; *Vucetich, 2012*). A higher estimate by *Gude et al. (2012)* has been questioned because of seeming errors in calculations (*Vucetich, 2012*), so their higher estimate needs replication or correction. We use the preceding meaning of sustainability, not the other meaning of sustain suggesting a wolf population can withstand 1 or 2 years of higher rates of mortality before extirpation. Our justification apart from the literature comes from the Wisconsin DNR itself, using the *Adams et al. (2008)* estimate in prior wolf-hunting plans (*Natural Resources Board, 2012*; *Natural Resources Board, 2014*), citation of those quota plans in 2021 (*Natural Resources Board, 2021a*), and explicit mention of using a 24% threshold on 15 February 2021 (*Natural Resources Board, 2021b*). Evaluating sustainability of natural resource uses demands long-term data, so here we only discuss the 1-year outcome in light of the objectives. Nevertheless, we can evaluate the state objective scientifically because we have official hunt statistics, official population estimates, and relevant, peer-reviewed scientific models. Namely, the wolves of Wisconsin were subject to two recent modeling efforts. First, models of population growth were built that took into account loosening of ESA protections as announced on 3 November 2020 (*Chapron & Treves, 2016a*; *Chapron & Treves, 2016b*; *Chapron & Treves, 2017a*; *Chapron & Treves, 2017b*); note we use 3 November from the Federal Register for consistency with prior studies (*Chapron & Treves, 2016a*; *Santiago-Ávila, Chappell & Treves, 2020*). Also, individual survival models used time-to-event analyses to estimate cryptic poaching in competing risk frameworks (*Santiago-Ávila, Chappell & Treves, 2020*). These allow us to estimate population change in a single year and increments in human-induced mortality following delisting and through the wolf-hunt period. The serendipitous combination of population estimates, hunter-killed totals, and models of the individual and population-level effects of reducing ESA protections make this case unique to our knowledge.

Another feature of the Wisconsin case that makes it relevant beyond that State are the subnational governance issues involved. The DNR was not alone in deciding or designing the state wolf hunt. A local court, the legislature, and the Natural Resource Board (NRB), which is a commission overseen by both the executive and the legislature, all had a say in the February 2021 wolf-hunt timing, methods, and quota (Material S1). Therefore, the Wisconsin case study may provide readers from other regions with insights into the checks and balances across three independent branches of a democratic government.

Here we evaluate whether the state attained its objective "…to allow for a sustainable harvest that neither increases nor decreases the state's wolf population…", by modeling population change after the State of Wisconsin issued 2,380 permits, intending to kill 119 wolves (https://dnr.wisconsin.gov/newsroom/release/41071, accessed 24 March 2021), but resulting in permitted kills of 218 wolves in <3 days (*Wisconsin, 2021a*).

## MATERIALS AND METHODS

We used official population estimates since April 2017 as the population grew from 925–1,034 minimum counts (Material S1) to estimate the population in April 2021. We began with population estimates and dynamics since April 2017, which represents the
most recent 4 years of wolf population growth after the last wolf-hunt in December 2015 (*Wisconsin, 2021b*). Therefore, we assume similar population dynamics, such as density-dependence, as observed in 2017–2020. We also assume the effects of that prior wolf-hunt had worked themselves out of the population dynamics preceding the wolf-hunt of February 2021. Some readers may be interested in seeing a 1-year population change model that allows for density-dependence or compensatory effects on mortality, reproduction, recruitment, or migration. In Material S2, we explain why a population model without such non-linear effects is the more conservative model.

We used three conservative scenarios for estimating population change. Our precautionary approach is to begin with the minimum bound of the April 2020 estimate by the State in its wolf population census. Our approach is precautionary because loners and transients contribute little to population growth or the total size of the population and few if any packs have been missed in previous years. Also, the minimum count of 1,034 wolves in 256 packs is consistent with long-term average pack sizes of approximately four wolves (*Wydeven et al., 2009*). Moreover, the state used 1034–1057 (SM1 Figure 2) and analogies to previous wolf-hunts that used the same wolf census method when the state recommended its quota for February 2021 (*Wisconsin, 2021c*).

The first scenario, which we label HIGH, uses the average growth estimated by the state during periods of strict ESA protection 2017–2020 $(N_{t+1} - N_t)/N_t = +3.8\%$, and accounts for mortality additional to background levels found during those years to account for the delisting period from 3 November 2020 to 14 April 2021. Specifically, we deduct additional deaths expected during periods without ESA protection from a recent peer-reviewed model of individual survival as policies changed.

Recent quantitative models predict that cryptic poaching—illegal killing in which perpetrators conceal evidence (*Liberg et al., 2012*)—rises significantly for endangered wolves when wolf-killing or removal from the wild, mostly by government agents, is legally permitted (*Santiago-Ávila, Chappell & Treves, 2020*; *Louchouarn et al., 2021*). The latter two recent models used independent datasets to estimate mortality and disappearance of marked wolves from the date of collaring (mainly VHF radio transmitters) until death or disappearance, using individual-level, time-to-event analyses to compare periods of strict ESA protection to periods of reduced protection during which time wolf-killing or removal of wild wolves to captivity was liberalized (*Santiago-Ávila, Chappell & Treves, 2020*; *Louchouarn et al., 2021*). The rationale for assigning most additional disappearances of radio-collared wolves to cryptic poaching follows discussions in those papers and others (*Treves et al., 2017*; *Agan, Treves & Willey, 2020*; *Treves et al., 2017*), which we summarized in Material S2, after describing depensatory mortality. The latter works improved upon earlier efforts (*Olson et al., 2015*; *Stenglein et al., 2015*), as did (*Stenglein, Wydeven & Deelen, 2018*), but those we use here also improved by explicitly accounting for radio-collared wolves that disappeared as a function of the length of time wolves were exposed to policy periods that reduced ESA protections (*Santiago-Ávila, Chappell & Treves, 2020*). Unregulated and often undocumented illegal killing (poaching) exceeded legal, reported wolf-killing in every population studied thus far (*Treves et al., 2017*; *Adams et al., 2008*; *Liberg et al., 2012*; *Agan, Treves & Willey, 2020*). Therefore, it is
essential to accurate monitoring and quota-setting that prudent managers consider these additional deaths and count all mortality, or at least all anthropogenic mortality, when planning and communicating public hunting seasons.

The second scenario, which we label MODERATE, uses the minimum growth estimated by the state in those years $(N_{t+1} - N_t)/N_t = -2.2\%$. Using the minimum population growth observed in the past 4 years is consistent with a precautionary approach, the findings for a population-level model of all wolves in Wisconsin and Michigan from 1995–2012 (*Chapron & Treves, 2016a*; *Chapron & Treves, 2016b*). Those studies report that periods of liberalized wolf-killing were associated with an unidentified and unreported source of mortality that slowed population growth, independent of legal killing, by 4–6% annually. These studies resisted quantitative and qualitative challenges without published support for alternative hypotheses of density-dependence on mortality (*Chapron & Treves, 2017a*; *Chapron & Treves, 2017b*; *Stien, 2017*; *Pepin, Kay & Davis, 2017*; *Olson et al., 2017*). Furthermore, social scientific data corroborated the population-level findings with independent datasets (*Browne-Nuñez et al., 2015*; *Hogberg et al., 2015*) and the authors' own findings (*Treves, Naughton-Treves & Shelley, 2013*; *Treves & Bruskotter, 2014*). This scenario also deducted additional wolf deaths as in the HIGH scenario.

Finally, for the third, LOW scenario, we took the minimum population growth observed in years of full ESA protection (−2.2%) and subtract another 5%, for a final decrement of −7.2%. The LOW scenario, adjusts the observed minimum growth downward by 5% $(N_{t+1} - N_t)/N_t = -7.2\%$, but does not add the additional mortality because that might double-count the effect of reduced protections after delisting on 3 November 2020.

## Assumptions

Our estimates contain a set of assumptions, all of which we aimed to make conservatively, so our outputs are minimum estimates of deaths and maximum estimates of population size.

We report only the increment in deaths and disappearances after delisting, i.e., those that we estimate would have survived had delisting not proceeded. We use these as increments in mortality for the HIGH and MODERATE scenarios only. The lower estimate for additional deaths and disappearances comes from wolves in Wisconsin from 1980–2012 (*Santiago-Ávila, Chappell & Treves, 2020*). The higher estimate for Mexican gray wolves, in New Mexico and Arizona, is more certain because of more intensive monitoring of a greater proportion of the population (*Louchouarn et al., 2021*). Therefore, the Wisconsin estimates are conservative among available estimates of cryptic poaching increments.

As summarized in Material S2, when we estimate additional wolf deaths and disappearances after delisting, we assume those wolves are lost to the Wisconsin population. Studies in at least four populations found that the vast majority of radio-collared wolf disappearances are earlier than would be expected from battery or mechanical failure (*Treves et al., 2017*; *Santiago-Ávila, Chappell & Treves, 2020*; *Liberg et al., 2012*; *Louchouarn et al., 2021*; *Agan, Treves & Willey, 2020*; *Treves et al., 2017*). We are aware of no evidence of a mechanism by which mechanical failure rates would
increase in association with a liberalized killing period. Further, the Scandinavian studies that first described cryptic poaching used genetics to confirm the disappearance of known wolves, and later associated those rates to policies, concluding that missing wolves no longer moved on the landscape, as opposed to eluding monitoring (*Liberg et al., 2012*; *Liberg et al., 2020*), but see our qualms about their inferences about policy effects (*Treves, Louchouarn & Santiago-Ávila, 2020*). Indeed, migration into, rather than out of, regions that experienced high rates of legal and illegal wolf-killing seems more likely. In the Alaskan gray wolf study widely used to identify a sustainability threshold for wolf-killing (*Adams et al., 2008*), the authors reported >75% of human-caused mortality was caused by intentional, unregulated hunting, and that the off-take was unsustainable without large amounts of immigration.

Also, we assumed no super-additive mortality per capita of legal kills, as reported or inferred for exploited wolf populations (*Creel & Rotella, 2010*; *Vucetich, 2012*), because we assume our estimates of cryptic poaching model some super-additivity. This is conservative because failed pregnancies, litter loss, and unreported deaths of uncollared wolves that might accompany and follow the hunting and poaching would not have been captured in the individual models that used marked adult wolves only. Non-radio-collared wolves succumbed to all deaths at higher rates than radio-collared wolves in Alaska (*Schmidt et al., 2015*), and in Wisconsin (*Treves et al., 2017*). Possibly some poachers are deterred by the threat of prosecution if they kill a collared animal (*Persson, Rauset & Chapron, 2017*). In sum, estimates of incremental deaths and disappearances in the HIGH and MODERATE scenarios are likely to under-estimate deaths.

Next we assumed permitted wolf-killing will have similar effects on the wolf population and on would-be wolf-poachers as that estimated from 2003–2012, during which time government agents were primarily responsible for wolf-killing and no public hunts were held. This is conservative given the 2021 wolf-hunt killed more wolves than in past periods (*Chapron & Treves, 2016a*; *Wisconsin, 2021b*), and did so with unprecedented methods (e. g., snowmobile chase, night-time, hounds, traps) in a very rapid timeframe. It would be plausible to assume rapid, efficient poaching also, but we do not.

Also, we assume all growth occurs prior to delisting because pups recruited into the population in November are treated as adults for purposes of census (*Fuller, 1989*). Relatedly, we assume that wolves alive on 15 April 2020 began their exposure to hazards at that time, rather than considering their full time alive as adults, for which we have no data. This is conservative because (1) the cumulative incidence (rather than the instantaneous hazard) of mortality increases with monitoring time naturally, and (2) the difference between the cumulative incidence functions for each protection period (Fig. 1) increases with monitoring time beyond our study period (t = 365) (*Santiago-Ávila, Chappell & Treves, 2020*).

Finally, we did not use unpublished, preliminary, unverified estimates provided by the DNR in April 2021 that 17 out of 50 collared wolves disappeared prior to or during the 2021 wolf-hunt and another 7 were killed by hunters (Materials S1, Fig. 1). Had we uncritically used those figures for deaths and disappearances of the entire wolf population, our estimate of wolf mortality would have been 48% and the associated wolf population

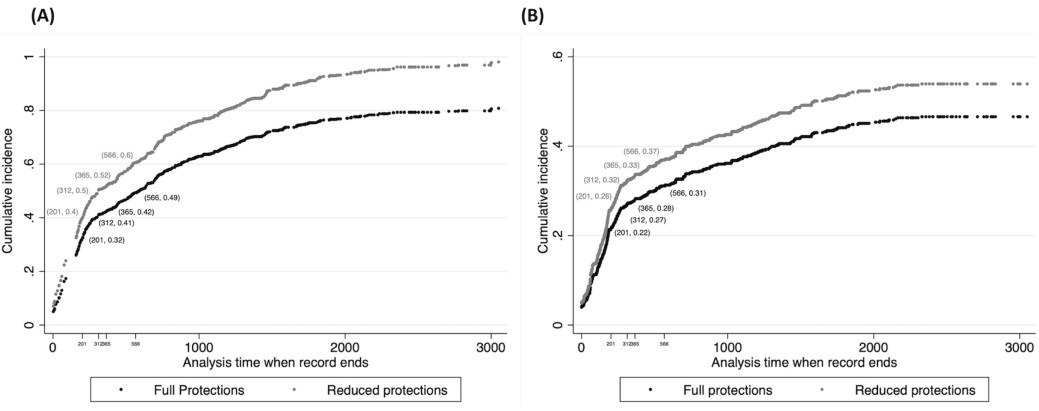

**Figure 1 Cumulative Incidence of endpoints by protection period.** Cumulative incidence functions (CIFs) for 499 monitored, adult wolves in Wisconsin during two policy periods (gray: reduced ESA protections; black: full ESA protections) for all deaths and disappearances (Panel A: $n$ = 499), and disappearances only (Panel B: $n$ = 243) from 1979–2012. Coordinates ($x$, $y$) represent the cumulative incidence or proportion of monitored wolves experiencing an endpoint (y-axis); showing all deaths in (A) or all disappearances in (B), over time (x-axis) in days. Time zero is set to 16 April 2020, a conservative step because death or disappearance increases with time, by definition. CIFs modeled with semi-parametric Fine-Gray models (*Santiago-Ávila, Chappell & Treves, 2020*). The first period of 201 days runs from 15 April 2020 to 3 November 2020 when delisting was announced in the Federal Register (*USFWS, 2020*) and the period of reduced ESA protection began. Day 312 marks the start of the Wisconsin wolf-hunt on 22 Feb 2021, and day 365 marks the end of the wolf-year on 14 April 2021. Finally, day 566 marks the approximate start date of the putative, next wolf-hunt, to illustrate further increases in the CIFs of mortalities and disappearances. We used the increment between the period of full ESA protection (black markers) to the corresponding value on the upper curve of reduced ESA protection (gray markers) to estimate the additional wolves deducted from the population by any endpoint (A) or LTF (B) which we predict would have survived if delisting had not occurred.

decline would have been much greater. But those data are unverified currently and, as noted above, collared wolves suffer different mortality hazard than uncollared ones in Wisconsin and elsewhere.

The formula we use for all three scenarios is Eq. (1)

$$N_{2021} = (N_{2020} \cdot r) - 218 - E \tag{1}$$

where $N_{2020}$ = 1,034, $r$ varies by scenario as +0.038 (HIGH), −0.022 (MODERATE), or −0.072 (LOW) respectively, and 'E' refers to additional wolves dead due to reduced ESA protections, calculated using the cumulative incidence functions (CIFs, Fig. 1A) for all endpoints during a period of liberalized wolf-killing from (*Santiago-Ávila, Chappell & Treves, 2020*), but set to zero for the LOW scenario. CIFs by policy periods for all endpoints and LTF (Figs. 1A, 1B) were calculated using semi-parametric Fine-Gray models, with data from 513 monitored, adult wolves (1979–2012) (*Santiago-Ávila, Chappell & Treves, 2020*).

We also estimate the proportion of all additional mortalities due to cryptic poaching, using the difference in CIFs for Radio-collared wolves lost to follow-up, in the two types of policy periods (Fig. 1B), divided by the same difference in the CIFs of all endpoints (Fig. 1A) at day 365 (15 April 2021).

**Table 1  Population and extra mortality estimation in scenario HIGH that assumes annual growth +3.8% by Apr 2021.**

| Timeline of wolf population changes | N | Individuals dead and disappeared | |
| --- | --- | --- | --- |
| | | Additional, due to reduced ESA protections[*] | Notes |
| **15 April 2020 in 256 packs, Day 0** | 1,034 | | We assume wolves begin monitoring on this date |
| **Expected by 2 Nov 2020, Day 201-** *REDUCED PROTECTION PERIOD BEGINS ON 3 NOV 2020* | 1,073 | 97 | Nov 3-Feb 21 (Days 202–312, 111 day interval): Liberalized wolf-killing period cumulative incidence as a relative increment of +0.09 for all endpoints relative to baseline of strict ESA protection |
| **Expected by 24 Feb 2021, Day 315-** *END OF WOLF-HUNT* | 759 | 218 | Legal kills during wolf-hunt Feb 22–24 (3 days) |
| **Expected by 15 Apr 2021, Day 365** | 751 | 8 | Feb 22-Apr 14 (Days 313–365, 51 day interval): Liberalized wolf-killing period cumulative incidence as a relative increment of +0.01 for all endpoints relative to baseline of strict ESA protection |

Note:
[*] Source for all cumulative incidences is *Santiago-Ávila, Chappell & Treves (2020)*.

**Table 2  Population and extra mortality estimation in scenario MODERATE that assumes annual change −2.2% by Apr 2021.**

| Timeline of wolf population changes | N | Individuals dead and disappeared | |
| --- | --- | --- | --- |
| | | Additional, due to reduced ESA protections[*] | Notes |
| **15 April 2020 in 256 packs, Day 0** | 1,034 | | We assume wolves begin monitoring on this date |
| **Expected by 2 Nov 2020, Day 201-** *REDUCED PROTECTION PERIOD BEGINS ON 3 NOV 2020* | 1,011 | 91 | Nov 3-Feb 21 (Days 202–312, 111 day interval): Liberalized wolf-killing period cumulative incidence as a relative increment of +0.09 for all endpoints relative to baseline of strict ESA protection |
| **Expected by 24 Feb 2021, Day 315-***END OF WOLF-HUNT* | 702 | 218 | Legal kills during wolf-hunt Feb 22–24 (3 days) |
| **Expected by 15 Apr 2021, Day 365** | 695 | 7 | Feb 22-Apr 14 (Days 313–365, 51 day interval): Liberalized wolf-killing period cumulative incidence as a relative increment of +0.01 for all endpoints relative to baseline of strict ESA protection |

Note:
[*] Source for all cumulative incidences is *Santiago-Ávila, Chappell & Treves (2020)*.

We do not attempt to model population change from 15 April 2021–November 2021 when the next wolf-hunt is putatively planned because there are too many uncertainties about reproduction, legality, and planning processes. A lack of transparency about state wolf data from 2013–2015 prevents independent scientific scrutiny of past regulated hunting (*Santiago-Ávila, Chappell & Treves, 2020*; *Treves et al., 2017*).

# RESULTS

We predict the state population by 15 April 2021 will stand at a maximum possible number of wolves of 695–751 wolves (scenarios: LOW 742, MODERATE 695, HIGH 751) (Tables 1, 2). This represents a minimum of a 27–33% decrease in 1 year. We emphasize

that is a minimum and the population size is a maximum because of the many conservative methods we used.

We estimate that in addition to the 218 wolves reported killed during the wolf-hunt, 98–105 wolves died since 3 November 2020 that would have been alive had delisting not occurred. Of these 56–63% (55–58 wolves) at a minimum would have been killed through cryptic poaching. Therefore, the addition of cryptic poaching and wolf-hunting in Wisconsin after 3 November 2020 seems to have augmented human-caused mortality by approximately 30% (320 of 1,034–1,071) over pre-delisting levels.

## DISCUSSION

We report the expected additional wolf mortality and population reduction in the aftermath of U.S. federal removal of endangered species protections followed by one state's swift adoption of a policy for liberalized wolf-killing, including permitted, public hunting, trapping, hounding, and snow-mobile pursuit by day and night. We estimate the incremental addition of at least 98–105 wolf deaths prompted by removing protections, of which cryptic poaching would comprise the majority, in addition to the hunting deaths.

We estimate a population reduction of at least 27–33% in 1 year, which contradicts the expectation by the state wildlife agency that there would be no reduction in the wolf population. Moreover, our estimates are strict minima for actual reductions in the population, so our population estimate is a maximum conceivable under the most conservative assumptions. The reality is probably a greater reduction and a lower population count as of writing.

If the second planned wolf-hunt in November 2021 (Material S1) were cancelled, we predict the state wolf population could rebound in 1–2 years. However, there are preliminary indications from the state Natural Resource Board that another wolf-hunt with a similar or higher quota will be advocated by some on the board (Material S1). Proponents for such point to the 1999 population goal for wolves of 350 individuals in late winter. We have shown that number is a value judgment by a few individuals not a scientifically sound target (Treves et al., 2021). Therefore, the adequacy of state regulatory mechanisms seems fragile, for reasons detailed in SM1 for those interested in policy background. The frailty of regulatory mechanisms can be summarized as follows:

1. The intervention of numerous branches of the state government (Material S1)
2. A Wisconsin statute which mandates a hunt in the event of federal delisting, rather than granting discretion to the DNR (Material S1)
3. Various disparate estimates of the population size, the hunter take, poaching, and resilience that have been espoused by officials and the public (Material S1)

In sum, the state wildlife agency (DNR) did not meet its explicit objectives of no change in the wolf population, still being advocated by that agency as of writing (https://dnr.wisconsin.gov/topic/hunt/wolf/index.html, accessed 16 April 2021). The facts of hunters over-shooting the quota by 83% before the DNR could close zones, of the Natural Resource Board over-ruling the DNR's more cautious permit number, the legislature mandating

a hunt, a county court ordering a hunt on very short notice, and an appeals court declining to review that decision (Material S1), all speak to problems with different branches intervening to reduce the discretion of the wildlife agency. That loss of discretion by the ostensible expert managers itself raises serious questions about the adequacy of regulatory mechanisms to prevent wolves becoming endangered again. It also leads us to recommend reform of trustee duties in the state and perhaps others with unclear responsibilities and unclear divisions between decision-making and implementation functions.

## CONCLUSIONS

For jurisdictions elsewhere, we caution that science may play little role in wolf politics where the animal has become a symbol for political rhetoric and a symbol of cultural divisions (*Nie, 2003*). However, science only reveals past, present or future conditions, not what we humans ought to do.

Proponents of wolf-killing argued that the state population goal of 350 wolves demands such swift reductions (Material S1), but evidence suggests that goal is a value judgment by a few individuals that was treated as if it were an output of a scientific model (*Treves et al., 2021*). Moreover, the model used suffers from scientific flaws, so its assumptions and predictions are dubious (*Treves et al., 2021*). Nevertheless, the goal was reaffirmed in February 2021 (Material S1). Furthermore, the state did not collect wolf carcasses for aging or detection of alpha females by placental scars, as is fairly standard for scientific studies, e.g., (*Stark & Erb, 2012*)—see Material S1 for tribal involvement in such analyses. This type of scientific information is indispensable for science-based management. Without it, illegal wolf-killing is more difficult to detect, the age and reproductive class of hunter-killed wolves is likely imprecise (*Treves et al., 2017*), and the breeding status and hence reproductive performance for the following year cannot be estimated accurately.

Likewise, state plans for another hunt raise questions about sustainability. Although one subnational jurisdiction may not predict another, doubts about sustainable wolf-killing and misuse of scientific information have been raised previously for several other governments (see *Creel et al. (2015)* and *Chapron et al. (2013)*, respectively). Therefore, we find our case is not unique, and provides insights for other jurisdictions. Similar wolf-killing might be replicated elsewhere when subnational jurisdictions in the USA and EU regain authority for controversial predators. Federal governments in both regions should recognize that loosening protections for predators, and perhaps other controversial species, opens the door for antagonists (*Treves & Martin, 2011*; *Brown & Samuels, 2021*) to kill large numbers in short periods, legally and illegally. The history of political scapegoating of wolves (*Chapron et al., 2013*; *Chapron & Lopez-Bao, 2014*) may repeat itself. Elsewhere, we have shown that the response should not be to allow more wolf-killing under the misguided concept of blood buys goodwill or 'tolerance killing' (*Chapron & Treves, 2017b*; *Santiago-Ávila, Chappell & Treves, 2020*; *Louchouarn et al., 2021*; *Treves & Bruskotter, 2014*).

Federal decision-makers might consider different classifications that make predators protected non-game, or states should prove themselves capable of reducing poaching to a stringent minimum for a 5-year post-delisting monitoring period. Alternately, federal governments might address upgrades to federal laws regardless of species classifications. Given the importance of predators in restoring ecosystem health and function (*Estes et al., 2011*) and of non-anthropocentric wildlife trusteeship (*Treves, Santiago-Ávila & Lynn, 2018*; *Santiago-Avila, Lynn & Treves, 2018*; *Santiago-Ávila, Treves & Lynn, 2020*), we also recommend instead that transparent legal standards of trusteeship be used to manage wildlife (*Bruskotter, Enzler & Treves, 2011*; *Treves et al., 2018*), not the vagaries of opaque electoral politics and interest group lobbying (*Treves et al., 2017*). Moreover, our recommendation conforms to global goals for the preservation of nature.

### Funding
The University of Wisconsin-Madison supported the salaries of Francisco J. Santiago-Ávila, Karann Putrevu, and Adrian Treves. The funders had no role in study design, data collection and analysis, decision to publish, or preparation of the manuscript.

### Grant Disclosures
The following grant information was disclosed by the authors:
The University of Wisconsin-Madison.

### Competing Interests
The authors declare that they have no competing interests.

### Author Contributions
- Adrian Treves conceived and designed the experiments, performed the experiments, analyzed the data, prepared figures and/or tables, authored or reviewed drafts of the paper, and approved the final draft.
- Francisco J. Santiago-Ávila conceived and designed the experiments, performed the experiments, analyzed the data, prepared figures and/or tables, authored or reviewed drafts of the paper, and approved the final draft.
- Karann Putrevu analyzed the data, authored or reviewed drafts of the paper, and approved the final draft.

### Data Availability
The data are available in the Supplemental Files.

### Supplemental Information
Supplemental information for this article can be found online at http://dx.doi.org/10.7717/peerj.11666#supplemental-information.

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
