# Peer review of "Quantifying the effects of delisting wolves after the first state began lethal management"

_PeerJ, doi:10.7717/peerj.11666_

## Round 0.1 · original submission · Major Revisions

Dear Adrian:

We expected this to be controversial. As you can see, one reviewer recommends major revision, the other rejection.

In preparing your revision, it will be essential to address all the concerns raised with particular care. For example, the penultimate comment in the "reject" reads:

"Line 189: there is no substance to these claims of cryptic poaching and excess human-cased mortality. Pure speculation."

You will need to respond to this directly in the manuscript itself and presenting your arguments for what you claim. I know that for rare, threatened species, data are few and we must often make judgments on inadequate information.

I will certainly send your revision out for further review. It may not convince the second reviewer, in which case (s)he may well choose the option to publish a response. I will give such a response considerable deference. That means you should write your revision defensible, expecting it to be criticised.

Reviewer 1 ·

Basic reporting

This paper is clearly written with appropriate structure and citations. The results are relevant to the hypotheses and the inferences are presented clearly.

This paper analyzes existing data on the size of the Wisconsin wolf population, reported offtake by human hunters, and rates of disappearance of radiocollared wolves. These data are combined in a very simple population model to assess whether the policies implemented by wildlife management agencies and the state government after delisting from the ESA are likely to achieve the stated goal of keeping population size constant. The authors use cumulative incidence functions fit to disappearance rates of radiocollared wolves in periods with and without hunting to demonstrate that increased legal hunting is associated with a substantial increase in mortality rates due to other causes, including illegal killing. Simple models that begin with the reported initial population size, allow it to grow at empirically measured rates (one high and one low) and subtracting out the number of legal and additional mortalities shows that the policy is likely to cause a substantial decrease in population size in a single year.

The strength of this paper is that the results just outlined are presented in a succinct and convincing manner. I find that the data are used appropriately, the methods and assumptions are stated clearly, and the inferences are supported by the data. One caveat is that the expected population sizes from these models are indeed expected, rather than demonstrated, but this could be viewed also as a strength, because the expectation (and its logic) is laid out proactively and can be tested.

The weakness of this paper is in the framing, which is often presented in language that I feel is likely to be inflammatory in a way that is not necessary, and indeed may interfere with productive evaluation of competing hypotheses about the mechanisms underlying the policies that are being evaluated. Even in cases in which I personally agreed with a sentiment being expressed (implicitly or explicitly), I often felt it might be framed in a more even-handed manner.

Experimental design

The central question is well defined, and the methods used to assess the hypothesis were appropriate. This work addresses an important knowledge gap, about the consequences for population dynamics of an abrupt change from full protection under the Endangered Species Act to high levels of legal recreational killing.

Validity of the findings

The underlying data are presented clearly and explained appropriately. The analysis is clear and appropriate. The conclusions are stated clearly and summarized appropriately in the Tables and Figures.

Additional comments

This paper analyzes existing data on the size of the Wisconsin wolf population, reported offtake by human hunters, and rates of disappearance of radiocollared wolves. These data are combined in a very simple population model to assess whether the policies implemented by wildlife management agencies and the state government after delisting from the ESA are likely to achieve the stated goal of keeping population size constant. The authors use cumulative incidence functions fit to disappearance rates of radiocollared wolves in periods with and without hunting to demonstrate that increased legal hunting is associated with a substantial increase in mortality rates due to other causes, including illegal killing. Simple models that begin with the reported initial population size, allow it to grow at empirically measured rates (one high and one low) and subtracting out the number of legal and additional mortalities shows that the policy is likely to cause a substantial decrease in population size in a single year.

The strength of this paper is that the results just outlined are presented in a succinct and convincing manner. I find that the data are used appropriately, the methods and assumptions are stated clearly, and the inferences are supported by the data. One caveat is that the expected population sizes from these models are indeed expected, rather than demonstrated, but this could be viewed also as a strength, because the expectation (and its logic) is laid out proactively and can be tested.
The weakness of this paper is in the framing, which is often presented in language that I feel is likely to be inflammatory in a way that is not necessary, and indeed may interfere with productive evaluation of competing hypotheses about the mechanisms underlying the policies that are being evaluated. Even in cases in which I personally agreed with a sentiment being expressed (implicitly or explicitly), I often felt it might be framed in a more even-handed manner.

For example line 280 criticizes “the vagaries of opaque electoral politics and patronage lavished upon interest groups” to conclude that “the history of political scapegoating of wolves will likely repeat itself”. Line 247 states that “only a government that values evidence as a way to inform decisions would heed scientists. Although we have stressed the conservative estimation procedure we used, we fear decision-makers will not heed the cautions.” In these passages, the intent of actions by others is being stated as though it was known, as are the values on which the actions were based, along with the likelihood of future actions. I think it would probably be better to avoid statements about intentions and values in the context of the simple population data that were analyzed here.

I find the Methods and Results sections well written and without this problem, but I would suggest revision of the material outside lines 109-242 to allow the analysis and results to stand more simply on its merits. I agree with the authors that there are many individuals with strongly held biases in this matter. I think this analysis makes an important contribution to the assessment of policy actions taken rapidly after delisting to allow (or even promote) high levels of human caused mortality. I am nonetheless concerned with the framing.

Reviewer 2 ·

Basic reporting

Excellent English, clearly written. Although a substantial number of references are cited, the literature was selected and does not include any of the papers showing how wolf harvests can be sustainable at much higher levels than reported here. The "hypothesis" is not sound and fails on any criteria for an experiment, i.e., no replication, no randomization, and no controls.

Experimental design

There is no experimental design, only flawed population models that are pure speculation.

Validity of the findings

Fundamentally flawed.

Additional comments

Wolves were delisted by the Donald Trump administration, returning management authority to the states. Wisconsin was quick to open a hunt because wolf numbers were substantially above their management target. The response by hunters was extraordinary and within 3 days hunters had killed many more wolves than was intended by the state. Because of this, Tewes et al. make an argument that state jurisdictions cannot be trusted with sound wolf management and claim that a sustained population of wolves cannot be maintained without federal intervention. They make up some numerical examples that are hypothetical and fail to include structural features in a population model, e.g., density dependence, that are essential for sound management of sustainable harvesting. The authors clearly have an agenda to attack the State of Wisconsin for its wolf management program and imply that management must return to federal protection.
Line 21: data is plural and this sentence grammatically incorrect
Line 30: cryptic poaching is not estimated using any statistically reliable method and attributing extensive mortality to this unknown is not defensible.
Line 36: the data and modeling exercise certainly cannot “debunk” the state’s interpretation
Line 56: a number of jurisdictions in North America have sustained culling of wolves and these data are not evaluated.
Line 66: lethal removal of wolves in the northern Rocky Mountains and in Canada contradict this claim.
Line 75: this is not a robust case study. Instead it is a one off where the hunter kill was much higher than anticipated by the State, and resulted in much higher kill than anticipated by the State. Based on experience in the Rocky Mountains, the state had every reason to expect that they could achieve their harvest objective. Their mistake does not constitute a robust case study.
Line 90: somehow I find it peculiar that the media would be a more credible source of population data than the state responsible for management.
Line 103: given that the hunt was just last fall, to conclude that the number of wolves has been reduced substantially is simply not credible.
Line 130: the population growth scenarios are just made up and cannot be regarded as a defensible analysis.
Line 155: the authors do not have more recent estimates because the state prohibits harassing hunters. Very strange logic.
Line 189: there is no substance to these claims of cryptic poaching and excess human-cased mortality. Pure speculation.
Line 194: it is well established that young of the year have higher mortality than adults.

---

## Round 0.2 · accepted · Accept

Dear Adrian:

I am accepting this version of your manuscript without further changes.

I sent the current version to the previous reviewers. One replied quickly and agreed it should be accepted.

I too think you have answered this reviewer adequately. Of particular concern was some of the language you used originally. Both the reviewer and I feel you have changed your tone adequately.

The other, more critical reviewer, did not choose to review again. I think this other will not be satisfied, but that there is no point in endless anonymous debate with him/her. (S)he did suggest another reviewer, but I chose not to invite a third one, for that might generate unreasonable delays.

Much experience teaches that the best way forward is to bring the dissenting views into the open. PeerJ is a particularly good platform for doing this — it has fast turnaround, is open access, and in other ways provides a way for bringing debates into the open.

For instance, authors have the right to make the review history public. That includes my decisions.

That said, should either reviewer or indeed any other interested party wish to publish a critique of your paper, I would offer them much the same conditions. That is, you would get a chance to review any critique — whether they asked for you to not be a reviewer or not. Moreover, even were you to recommend rejection, I might accept a revision that made a good faith attempt to answer your concerns without an inevitable second review.

PeerJ's policy on this is that PeerJ doesn’t publish simple rebuttals or response papers, *unless* they are new standalone research papers in their own right (e.g. which reanalyse the data and come to a new conclusion). The policy on that is at https://peerj.com/about/policies-and-procedures/#critique-submissions. In our field, that covers a lot ground.

We both know your paper will generate controversy.

For the record: I have added my name to scientists' sign-on letters about wolves. Over the years, I have also been the plaintiff in several cases brought by the Center for Biological Diversity, typically on endangered species in Florida, where I have "standing." I do not have standing for wolves in Wisconsin.

Reviewer 1 ·

Basic reporting

My prior assessment was that this manuscript takes a simple but clear approach to assessing whether policies adopted by the state of Wisconsin after delisting of the wolf were likely to achieve a stated goal of maintaining constant population size. As before, I feel that the model's structure, assumptions and parameterization are clear and make use of public data in a useful manner. The question of identifying the consequences of dramatic and rapid changes in policy with changes in jurisdiction is important and timely. The additions and revisions in this version have considered and directly addressed prior comments.

Experimental design

As noted previously, the use of CIFs fit to data on disappearance rates is well justified, and the combination of these results with simple, direct estimates from agency reports of population size and growth rates is logically applied and well explained.

Validity of the findings

I find the revisions appropriate and well explained. All comments have been addressed, including some that seemed to me to be difficult to address because (in my opinion) they were highly critical without providing clear explanations or citations.